## METHODS & TECHNIQUES

# A simple, fast and inexpensive approach using *E. coli* to detect and estimate vitamin B$_{12}$ content in microbial extracts

**Katarzyna Hencel[1,2,*], Matthew J. Sullivan[1,2,*] and Alper Akay[1,2,*]**

## ABSTRACT

Vitamin B$_{12}$ is an essential micronutrient produced only by prokaryotes, and animals must acquire it from their diet. Vitamin B$_{12}$ is critical for the synthesis of methionine and propionyl-CoA metabolism. In humans, vitamin B$_{12}$ deficiency has been linked to many disorders, including infertility and developmental abnormalities. The growing trend towards plant-based diets and ageing populations increases the risk of vitamin B$_{12}$ deficiency, and, therefore, there is an increasing interest in understanding vitamin B$_{12}$ biology. Accurate approaches for detecting and quantifying vitamin B$_{12}$ are essential in studying its complex biology, from its biogenesis in Bacteria and Archaea to its effects in complex organisms. Here, we present an approach using the commonly available *E. coli* methionine auxotroph strain B834 (DE3) and a multi-well spectrophotometer to detect and estimate the levels of vitamin B$_{12}$ from biological samples at picomolar concentrations. We further show that our method is sufficient to reveal important differences in the production of vitamin B$_{12}$ from vitamin B$_{12}$-synthesising bacteria commonly found in the microbiome of wild *Caenorhabditis elegans* isolates. Our results establish a high-throughput and simple assay platform for detecting and estimating vitamin B$_{12}$ levels using the *E. coli* B834 (DE3) strain.

**KEY WORDS: Vitamin B$_{12}$, Cobalamin, *Caenorhabditis elegans*, *C. elegans*, *E. coli*, MetE, CeMbio**

## INTRODUCTION

Cobalamin (the natural form of vitamin B$_{12}$) is, structurally, the most complex vitamin. It is only produced by bacteria and archaea and requires more than a dozen enzymes for its biogenesis. In many organisms, cobalamin (vitamin B$_{12}$ hereafter) is essential for the function of two critical enzymes: methionine synthase, which regenerates methionine in the cells from homocysteine, and methylmalonyl-CoA mutase, which converts propionyl-CoA into succinyl-CoA. In humans, vitamin B$_{12}$ deficiency has been linked to multiple diseases, including anaemia, infertility, and developmental and neurological disorders (Mischoulon et al., 2000; Molloy et al., 2008). Although clinical levels of vitamin B$_{12}$ deficiency are rare (Green et al., 2017), the global increase in plant-based diets and ageing populations are linked to reduced vitamin B$_{12}$ uptake, which is considered a growing global health risk that necessitates further molecular and medical research into vitamin B$_{12}$ and its roles in human and animal physiology (Brouwer-Brolsma et al., 2015; Niklewicz et al., 2023).

Research on vitamin B$_{12}$ requires sensitive detection and quantification methods. In particular, previous studies in *Caenorhabditis elegans* have shown that vitamin B$_{12}$ affects animal development and fecundity in a dose-dependent manner (Bito et al., 2017, 2013; Watson et al., 2016). Therefore, accurate measurement of vitamin B$_{12}$ levels in animal diets would facilitate better analysis of the link between vitamin B$_{12}$ and development. Using vitamin B$_{12}$ auxotrophy in bacteria to quantify vitamin B$_{12}$ in biological samples has been a standard method since the 1940s when *Lactobacillus leichamnnii* was described by multiple groups as a suitable strain for vitamin B$_{12}$ quantification (Hoffmann et al., 1949, 1948; Kelleher and Broin, 1991; Skeggs et al., 1948). The assay has been used to this day and is readily available through commercial routes. However, *L. leichamnnii* has complicated growing conditions, including its response to thymidine in the absence of vitamin B$_{12}$ (Kitay et al., 1950).

Another bacterial strain commonly used for vitamin B$_{12}$ assays is *Salmonella typhimurium* mutants, which lack the vitamin B$_{12}$-independent methionine synthase, MetE (Raux et al., 1996). However, *Salmonella* strains grow under anaerobic conditions, often requiring additional apparatus. Another well-established assay uses the microalgae *Euglena gracilis var. bacillaris* (Hutner et al., 1949). Although considered more accurate, this assay takes 4 to 7 days to complete and involves complicated growth conditions.

Alternatively, methionine auxotroph strains of *Escherichia coli* can also be used for vitamin B$_{12}$ quantification (Chiao and Peterson, 1953; Davis and Mingioli, 1950; Harrison et al., 1951; Johansson, 1953). The advantages of utilising *E. coli* strains for vitamin B$_{12}$ assays include fast growth, which allows the assay to be performed overnight, and simple growth requirements, which do not involve specialised media preparation. However, there is limited information on the sensitivity and specificity of *E. coli*-based vitamin B$_{12}$ assays.

In addition to microbiological methods, several analytical methods have been developed for detecting and quantifying vitamin B$_{12}$ using liquid chromatography (Wongyai, 2000; Moreno and Salvadó, 2000), reverse-phase high-performance liquid chromatography (Heudi et al., 2006; Campos-Gimnez et al., 2008), and liquid chromatography coupled to mass spectrometry (Gentili et al., 2008; Chamlagain et al., 2015). Although some of these methods offer high sensitivity and lower detection levels, these techniques require high-end instruments and significant expertise in chromatography and mass spectrometry. In addition, the results from analytical approaches correlate well with microbiological assays (Campos-Gimnez et al., 2008).

Here, we present a high-throughput bacteria-based assay for vitamin B$_{12}$ detection and relative quantification using the

[1]School of Biological Sciences, University of East Anglia, Norwich, NR4 7TJ, UK.
[2]Centre for Microbial Interactions, Norwich Research Park, Norwich, NR4 7UJ, UK.

*Authors for correspondence (k.hencel@uea.ac.uk; matthew.sullivan@uea.ac.uk; a.akay@uea.ac.uk)

M.J.S., 0000-0003-2276-3132; A.A., 0000-0001-6825-4443

Biology Open

methionine-auxotroph *E. coli* B834 (DE3) strain. This approach allows for simple and time-efficient detection and relative quantification of vitamin $B_{12}$ content in biological samples. We further utilise the method to estimate the vitamin $B_{12}$ content of different bacterial species commonly found with wild isolates of the nematode *C. elegans*.

## RESULTS

### Analysis of *E. coli* B834 methionine and vitamin $B_{12}$ auxotrophy

*E. coli* has two methionine synthase enzymes: the $B_{12}$-dependent MetH and the $B_{12}$-independent MetE. *E. coli* B834 (DE3) has a *null* mutation in the *metE* gene, making the bacteria solely dependent on either methionine or vitamin $B_{12}$ supplementation. To assess whether *E. coli* B834 (DE3) is suitable for vitamin $B_{12}$ detection and quantification, we tested the growth of this strain in response to various vitamin $B_{12}$ and methionine supplementations. We confirmed that *E. coli* B834 (DE3) can grow only when methionine or vitamin $B_{12}$ is present in the media (Fig. 1A). We did not observe any significant difference in growth when $B_{12}$ was supplemented at concentrations ranging from 1 nM to 1000 nM, as determined by area under the curve analysis followed by one-way ANOVA with Holm-Šidák multiple comparison corrections (Fig. 1B).

### Using the growth of *E. coli* B834 to estimate vitamin $B_{12}$ levels

We subsequently sought to test the utility of using the growth of *E. coli* B834 (DE3) as a highly sensitive biological method for detecting vitamin $B_{12}$. To this end, we used M9 minimal media (devoid of methionine or vitamin $B_{12}$) and added vitamin $B_{12}$, supplemented at concentrations ranging from 0.00001 nM to 1 nM using 10-fold (Fig. 1C,D) and 2-fold (Fig. 1E,F) serial dilutions. Using this range, we determined that vitamin $B_{12}$ concentrations at and above 0.25 nM (250 pM) were sufficient to support the growth of the *E. coli* B834 (DE3) strain, as evidenced by a significant increase in area under the curve measurements throughout the growth period (Fig. 1F). Next, we prepared a standard curve of vitamin $B_{12}$ concentrations between 0 and 0.4 nM to determine the limit of detection and quantification for vitamin $B_{12}$ using the *E. coli* B834 (DE3) strain. Compared to the unsupplemented media control, the limit of detection is 0.05 nM (50 pM) (Fig. 2A,B), indicating that the growth of *E. coli* was detectable above background absorbance measurements. Increasing the carbon source from 0.4% glucose to 1.0% did not change the sensitivity of the growth assay (Fig. S1A,B). In summary, we have established that the growth of *E. coli* B834 (DE3) can be used to detect vitamin $B_{12}$ at concentrations as low as 50 pM and for relative quantification between concentrations of 50–200 pM.

### Using *E. coli* B834 (DE3) to estimate vitamin $B_{12}$ levels in biological samples

*C. elegans* is a well-established model organism for studying the function of vitamin $B_{12}$ during animal development and for understanding the molecular pathways related to vitamin $B_{12}$ (Bito et al., 2017, 2013, 2019; Bito and Watanabe, 2016; Watson et al., 2014). *C. elegans* exclusively feeds on bacteria, and its uptake of vitamin $B_{12}$ depends on the bacteria available in its environment as a food source. One such bacterium *C. elegans* feeds on in the wild is *Comamonas aquatica* DA1877, a known vitamin $B_{12}$ producer (Watson et al., 2014), which was isolated from soil (Shtonda and Avery, 2006). Mutations in the *cbiA* and *cbiB* genes, which code for cobyrinate a,c-diamide synthase and

adenosylcobinamide-phosphate synthase enzymes, respectively, prevent *C. aquatica* from producing vitamin $B_{12}$ (Watson et al., 2014). As a negative control for our assay, we generated an isogenic $\Delta cbiA\Delta cbiB$ mutant of DA1877 by deleting the *cbiA* and *cbiB* genes (Fig. S2A–C). To confirm that our $\Delta cbiA\Delta cbiB$ strain no longer produced vitamin $B_{12}$, we utilised our *E. coli* B834 (DE3) approach to test for the presence of vitamin $B_{12}$ in cell-free extracts from *C. aquatica*. Briefly, bacterial cells were lysed by boiling, and cell-free extracts were added to *E. coli* B834 (DE3) in media devoid of vitamin $B_{12}$ or methionine. Using this approach, we assayed the vitamin $B_{12}$ levels of wild-type *C. aquatica*, *C. aquatica* $\Delta cbiA\Delta cbiB$ and *E. coli* OP50, a different strain of *E. coli* commonly used as laboratory food for *C. elegans* but known to be a vitamin $B_{12}$ non-producer (Watson et al., 2014; Zimmermann et al., 2020). As predicted, cell-free extracts of wild-type *C. aquatica* DA1877 supported vitamin $B_{12}$-dependent growth of *E. coli* B834 (DE3), whereas the *C. aquatica* $\Delta cbiA\Delta cbiB$ mutant and *E. coli* OP50 did not (Fig. 2C,D). We further estimated the vitamin $B_{12}$ content in *C. aquatica* extracts by using 2-fold and 10-fold serial dilutions and comparing the relative growth of *E. coli* B834 (DE3) with *C. aquatica* extracts against growth with known concentrations of vitamin $B_{12}$ (Fig. S2D,E). Using the 2-fold and 10-fold dilutions combined with either the Gompertz-modelled Area under the Curve analysis (Fig. 2E) or the linear regression model (Fig. 2F) of the standard curve, we estimate the vitamin $B_{12}$ content of *C. aquatica* DA1877 to be approximately 25 nM and 27 nM per 1 OD unit (1 ml of culture at 1.0 $OD_{600nm}$) of bacteria, respectively.

Our assays with vitamin $B_{12}$, along with extracts from both vitamin $B_{12}$-producing and non-producing bacteria, provided proof of concept for our method to detect vitamin $B_{12}$ in complex biological samples by using the growth of *E. coli* B834 (DE3) as a proxy. Next, we applied this method to assess the vitamin $B_{12}$ content of 12 bacterial strains from the CeMbio collection, all of which were isolated from *C. elegans* found in the wild (Dirksen et al., 2020). Four bacterial strains, *Comamonas piscis* BIGb0172, *Pseudomonas berkeleyensis* MSPm1, *Pseudomonas lurida* MYb11 and *Ochrobactrum vermis* MYb71, were predicted to produce vitamin $B_{12}$ based on their genomic sequences and predicted metabolic pathway analyses (Zimmermann et al., 2020; Dirksen et al., 2020). Our analysis using the *E. coli* B834 (DE3) growth assay showed that *C. piscis* BIGb0172, *P. berkeleyensis* MSPm1, *P. lurida* MYb11, and *O. vermis* MYb71 are indeed vitamin $B_{12}$ producers, because cell-free extracts from cultures of these bacteria were capable of supporting the growth of *E. coli* B834 (DE3) in a manner that relied on vitamin $B_{12}$ (Fig. 3A,B). Among these, extracts from *C. piscis* BIGb0172 and *P. berkeleyensis* MSPm1 supported the highest growth, while *O. vermis* MYb71 showed reduced growth, indicating that there may be variation in the amount of vitamin $B_{12}$ produced by these bacteria. In contrast, supplementing *E. coli* B834 (DE3) with extracts from the other eight CeMbio strains led to a complete absence of bacterial growth (Fig. 3A,B).

In summary, we show that our application of *E. coli* B834 (DE3) growth in minimal media can be used for rapid and high-throughput detection and estimation of vitamin $B_{12}$ levels in biological samples.

## DISCUSSION

Vitamin $B_{12}$-dependent microorganisms are commonly used to detect and quantify vitamin $B_{12}$ in various formats (Hoffmann et al., 1949, 1948; Skeggs et al., 1948). Using *E. coli* *metE* mutants for this purpose offers numerous advantages, including their commercial availability, rapid growth, and simple growth requirements.

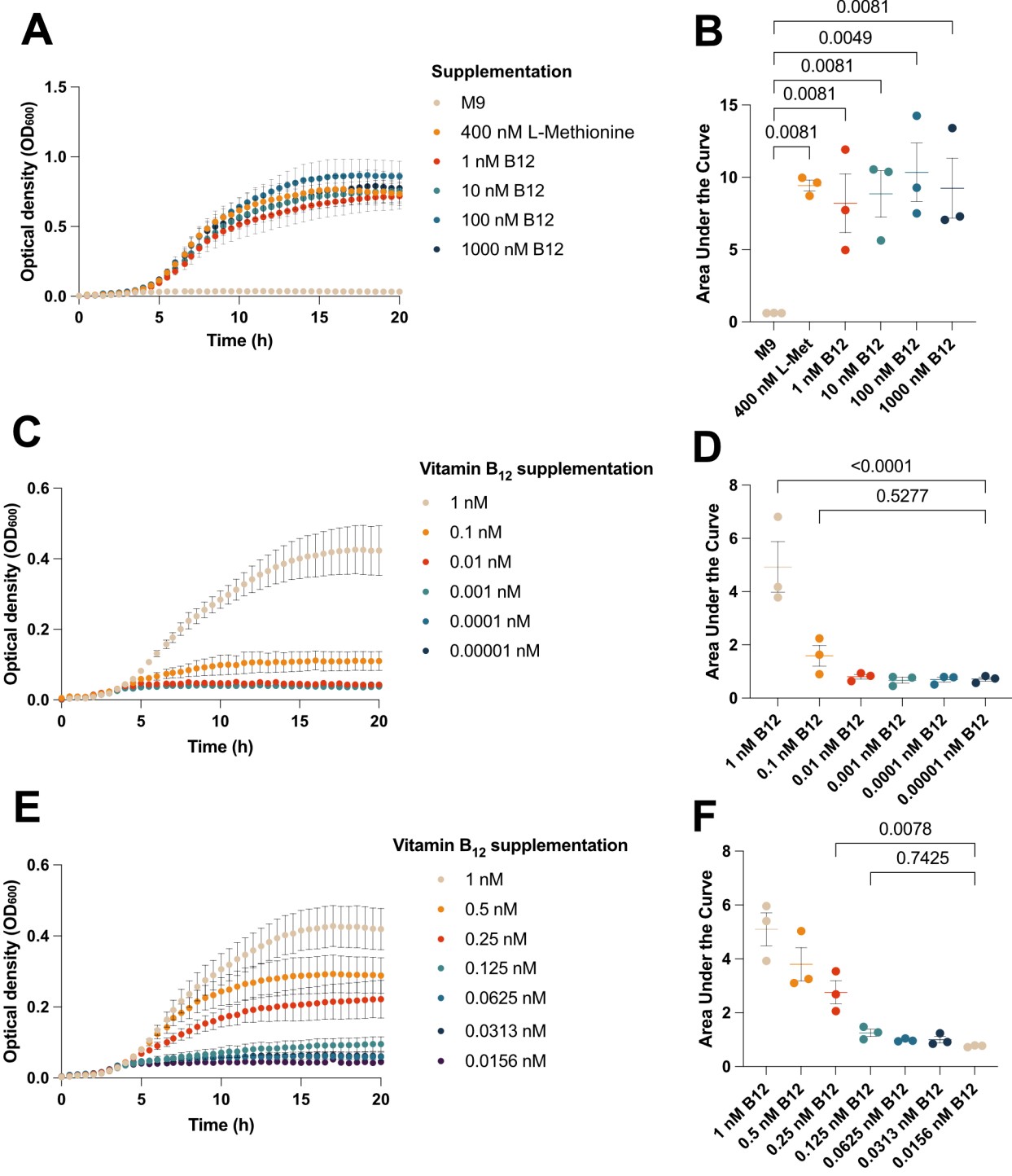

**Fig. 1. The growth of *E. coli* B834 (DE3) is dependent on methionine or vitamin B$_{12}$ and is titratable with vitamin B$_{12}$ concentration.** *E. coli* B834 was cultured in M9 minimal media, without or with supplemental methionine or vitamin B$_{12}$ (1 nM–1000 nM) as indicated (A). The growth conditions were compared by AUC analysis, followed by one-way ANOVA with Holm-Šidák multiple comparison corrections and *P*-values indicated (B). *E. coli* B834 growth with vitamin B$_{12}$ supplementation at 1 nM and using 10-fold (C) and 2-fold serial dilutions (E) to test a broad range of sub-nanomolar vitamin B$_{12}$ concentrations. The growth of *E. coli* B834 in the 10-fold and 2-fold were compared by AUC analysis followed by one-way ANOVA and Holm-Šidák multiple comparison corrections and comparisons are shown in D and F, respectively. The data points are plotted as mean±s.e.m. from three biological replicates derived from two technical replicates.

However, there is limited information on the sensitivity and reproducibility of *E. coli metE*-based vitamin B$_{12}$ assays. Here, we present a vitamin B$_{12}$ quantification assay using a readily available commercial strain of *E. coli* B834 (DE3) and widely used and inexpensive minimal media. The assay was developed in liquid culture using a 96-well plate format and a multi-well plate reader, allowing for reproducible analysis of many biological samples with a sensitivity as low as at 50 pM concentration.

The dependency on the growth of *E. coli* B834 (DE3) due to the presence of vitamin B$_{12}$ can be confounded by the presence of

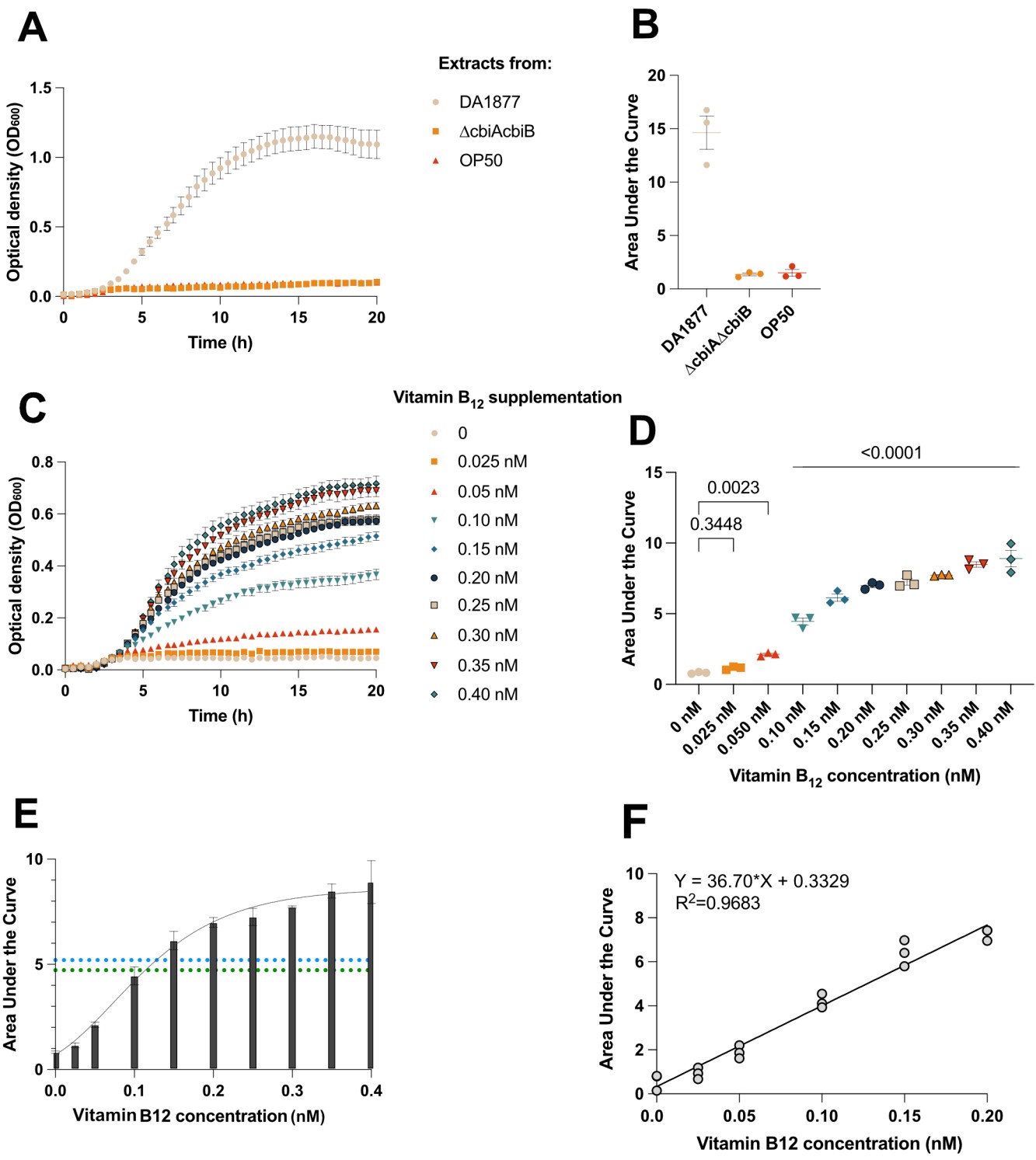

**Fig. 2. Use of *E. coli* B834 (DE3) growth to detect and estimate vitamin B$_{12}$ production by *C. aquatica*.** *E. coli* B834 (DE3) was cultured in M9 minimal medium, without or with supplemental vitamin B$_{12}$ at concentrations between 0.025–0.4 nM (A) and compared using AUC analysis coupled with one-way ANOVA with Holm-Šidák multiple comparison corrections (B). *E. coli* B834 culture was supplemented by bacterial cell-free extracts of *C. aquatica* DA1877, $\Delta cbiA\Delta cbiB$ *C. aquatica*, and *E. coli* OP50 (C) and growth in the presence of extracts were compared by AUC analysis coupled with one-way ANOVA with Holm-Šidák multiple comparison corrections (D). AUC data from vitamin B$_{12}$ standards in panel B were used for Gompertz model fitting, further employed for vitamin B$_{12}$ quantification [$Y=YM \times (Y0/YM)^\wedge(exp(-K \times X))$], where YM is the maximum AUC score, Y0 is the minimum AUC score, K determines the lag time (E). The dashed green line corresponds to the 4.70 AUC score of the $10^{-1}$ dilution of *C. aquatica* DA1877 extract, while the dashed blue line corresponds to the 5.20 AUC score of the 1:8 dilution of *C. aquatica* DA1877 extract, where 2 μl out of 50 μl of the 1 OD extract was used for quantification. In this model, YM=8.606, Y0=0.7062, and K=12.53. (F) Area under the curve analysis of *E. coli* B834 cultures supplemented with vitamin B$_{12}$ concentrations prepared from known standards. The line represents simple linear regression, and the equation and R$^2$ value are shown from three independent experiments. All data points are plotted with mean±s.e.m. from three biological replicates derived from two technical replicates.

Biology Open

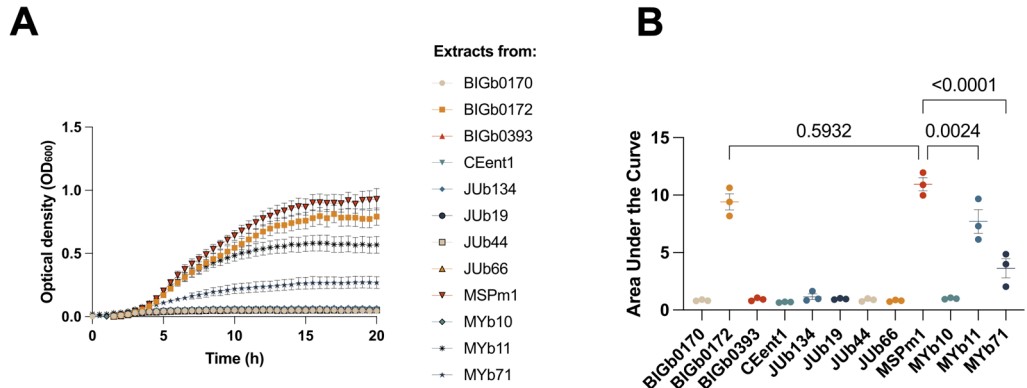

**Fig. 3. Detection and estimation of vitamin B₁₂ levels in bacterial isolates from wild *C. elegans*.** *E. coli* B834 (DE3) was cultured in M9 minimal media and supplemented with cell-free bacterial extracts of the indicated strains (A). Growth was compared using AUC analysis coupled with one-way ANOVA with Holm-Šidák multiple comparison corrections (B). The data points are plotted with mean±s.e.m. from three biological replicates derived from two technical replicates corrected for blank readings. Bacterial isolates are as follows: *Sphingobacterium multivorum* BIGb0170, *Comamonas piscis* BIGb0172, *Pantoea nemavictus* BIGb0393, *Enterobacter hormaechei* CEent1, *Sphingomonas molluscorum* JUb134, *Stenotrophomonas indicatrix* JUb19, *Chryseobacterium scophthalmum* JUb44, *Lelliottia amnigena* JUb66, *Pseudomonas berkeleyensis* MSPm1, *Acinetobacter guillouiae* MYb10 *Pseudomonas lurida* MYb11, *Ochrobactrum vermis* MYb71.

methionine, which may bypass the metabolic bottleneck caused by vitamin B₁₂ limitation in *metE⁻ E. coli* and could affect the specificity of our assay. However, previous studies conducted on the *E. coli* 113-3 strain, another methionine auxotroph, showed that methionine must be 50,000 times more concentrated than vitamin B₁₂ to hinder vitamin B₁₂ quantification using the *E. coli* assay, which was significantly higher than the levels found in the mammalian tissues tested (Chiao and Peterson, 1953). Similarly, we did not observe unexpected *E. coli* B834 (DE3) growth supported by extracts derived from known vitamin B₁₂ non-producers (Fig. 2C) or CeMbio collection strains, which were not predicted to produce vitamin B₁₂ (Fig. 3). However, we did not test a wider range of culture conditions, such as vitamin B₁₂ concentrations above 1 nM, but as highlighted in this work, we predict such levels would be sufficient for growth of the *E. coli* strain used in the present study. In addition, we have not performed a side-by-side comparison with other bacterial isolates used previously for microbiological assays of vitamin B₁₂. Therefore, in future work, we recommend additional testing of culture conditions and using a known vitamin B₁₂-deficient strain as controls. In

addition, the range of relative quantification is between 50 and 200 pM, which is small and should be used with caution, noting that samples may require dilution to be within this range. For absolute quantification of vitamin B₁₂, analytical methods using liquid chromatography and mass spectrometry should be considered as the validated approach.

Previous studies suggested that four strains in the CeMbio collection, *C. piscis* BIGb0172, *P. berkeleyensis* MSPm1, *P. lurida*, and *O. vermis* Myb71, are vitamin B₁₂ producers, based on the presence of vitamin B₁₂ biosynthetic pathway genes (Zimmermann et al., 2020). Our analysis provided experimental evidence to support this. Interestingly, despite confirming that all four isolates are vitamin B₁₂ producers, we note that the levels of vitamin B₁₂ likely vary significantly, with *P. berkeleyensis* MSPm1 and *C. piscis* BIGb0172 producing significantly higher levels of vitamin B₁₂ compared to *P. lurida* Myb11 and *O. vermis* Myb71. These differences in vitamin B₁₂ content could be important for the growth of *C. elegans* and other organisms that directly rely on bacteria for vitamin B₁₂.

## Table 1. Bacterial strains used in the study

| Strain name | Characteristics | Reference |
|---|---|---|
| *E. coli* B834 (DE3) | F⁻ *ompT hsdS*ᵦ(rₑ̄ mₑ̄) *gal dcm met* (DE3) | (Wood, 1966) |
| *E. coli* OP50 | *ura-, strR, rnc-,* (delta)*attB*::FRT-*lacI-lacUV5p*-T7 | (Brenner, 1974) |
| *E. coli* JKE201 | RP4-donor for bi-parental conjugation | (Harms et al., 2017) |
| *Comamonas aquatica* DA1877 | Wild-type strain | (Shtonda and Avery, 2006) |
| ALP121 | DA1877-derivative with *cbiB* deletion | This study |
| ALP122 | DA1877-derivative with *cbiA* and *cbiB* deletions | This study |
| | CeMbio collection | |
| *Sphingobacterium multivorum* BIGb0170 | Wild-type strain | (Dirksen et al., 2020) |
| *Comamonas piscis* BIGb0172 | Wild-type strain | (Dirksen et al., 2020) |
| *Pantoea nemavictus* BIGb0393 | Wild-type strain | (Dirksen et al., 2020) |
| *Enterobacter hormaechei* CEent1 | Wild-type strain | (Dirksen et al., 2020) |
| *Sphingomonas molluscorum* JUb134 | Wild-type strain | (Dirksen et al., 2020) |
| *Stenotrophomonas indicatrix* JUb19 | Wild-type strain | (Dirksen et al., 2020) |
| *Chryseobacterium scophthalmum* JUb44 | Wild-type strain | (Dirksen et al., 2020) |
| *Lelliottia amnigena* JUb66 | Wild-type strain | (Dirksen et al., 2020) |
| *Pseudomonas berkeleyensis* MSPm1 | Wild-type strain | (Dirksen et al., 2020) |
| *Acinetobacter guillouiae* MYb10 | Wild-type strain | (Dirksen et al., 2020) |
| *Pseudomonas lurida* MYb11 | Wild-type strain | (Dirksen et al., 2020) |
| *Ochrobactrum vermis* MYb71 | Wild-type strain | (Dirksen et al., 2020) |

**Table 2. Plasmids used in this study**

| Plasmid | Description | Reference |
| --- | --- | --- |
| pFOK | Suicide vector containing *sacB* and driven by p*tetA*; kan$^R$ | (Cianfanelli et al., 2020) |
| pAA46 | Derivative of pFOK containing up- and downstream fragments of the *cbiB*; kan$^R$ | This study |
| pAA47 | Derivative of pFOK containing up- and downstream fragments of the *cbiA*; kan$^R$ | This study |

In conclusion, our results establish a high-throughput, straightforward, and cost-effective method for detecting and estimating vitamin B$_{12}$ levels in biological samples. The simplicity, reproducibility, and sensitivity of the *E. coli* B834 (DE3) assay provide an important methodology for the research community working on vitamin B$_{12}$. Our discovery of varying vitamin B$_{12}$ levels in the wild *C. elegans* microbiome makes a compelling case for further investigation into how differences in bacterial metabolites impact animal development. Finally, we recognise that, as we have shown in the present study, growth-based approaches using *E. coli* may be applied to measure other metabolites of interest in a manner that is inexpensive and high-throughput; this area of microbiology should not be forgotten as a powerful functional approach in the biosciences.

## MATERIALS AND METHODS
### Bacterial strains, plasmids and culture media
All bacteria and plasmids are listed in Tables 1 and 2, respectively. Growth media used were M9 minimal salts medium (KH$_2$PO$_4$, 15 g/l NaCl, 2.5 g/l Na$_2$HPO$_4$, 33.9 g/l NH$_4$Cl, 5 g/l, 2 mM MgSO$_4$, 0.1 mM CaCl$_2$, 0.4% glucose unless otherwise stated), soya-rich medium (soya peptone 20 g/l, sodium chloride 5 g/l) or Lysogeny broth (LB; 10 g/l NaCl, 10 g/l tryptone, 5 g/l yeast extract). For *E. coli* B834, defective for *metE*, M9 medium was supplemented with 400 nM L-methionine as indicated. *E. coli* OP50, *C. aquatica* DA1877 and the isogenic Δ*cbiA*Δ*cbiB* mutant were grown in the soya-rich medium at 37°C at 180 rpm agitation for the extract preparation. All *E. coli* and *C. aquatica* derivatives were grown at 37°C at 180 rpm agitation, and strains from the CeMbio collection for the extract preparation were grown at 28°C at 180 rpm in the vitamin B$_{12}$-deficient soya-rich medium.

### *C. aquatica* DA1877 vitamin B$_{12}$-deficient mutant generation
Oligonucleotide primers (Table 3) with flanking 20 bp overhangs were designed to amplify upstream and downstream fragments from *cbiA* and *cbiB* of *C. aquatica* DA1877 using Benchling's Gibson Assembly Wizard. The amplified fragments were introduced to the pFOK suicide vector through Gibson Assembly and transformed into *E. coli* JKE201. All constructs were verified by Sanger sequencing, followed by conjugation with *C. aquatica* DA1877 on LB supplemented with 100 μM diaminopimelic acid (DAP) to support *E. coli* JKE201 growth. Transconjugants were selected onto LB agar containing 100 μg/ml kanamycin. At least three transconjugants were grown in LB medium for 4 h, followed by plating on no-salt LB agar plates (10 g/l tryptone, 5 g/l yeast extract, 15 g/l agar) supplemented with 20% sucrose and 0.5 μg/ml anhydro-tetracycline. Candidate colonies were screened for deletions through PCR and verified by Sanger sequencing. Mutation in *cbiB* resulted in the out-of-frame deletion of 178 amino acids, while *cbiA* mutation resulted in complete gene removal. The Sanger sequencing trace files are available as Supplementary File 1.

### Bacterial lysate preparation
Bacterial cultures for the *E. coli* B834 (DE3) assay were grown overnight and 1 OD unit (equivalent of 1 ml of culture with absorbance at 600 nm of 1.0) was centrifuged at 15,000 rpm for 1 min. The supernatant was removed, and the cells were resuspended in 50 μl of M9 minimal salts medium and boiled at 100°C for 15 min, as previously described (Ross, 1952). After boiling, lysates were centrifuged at 15,000 rpm for 1 min to remove debris, and the cooled supernatant was used as an extract for supplementation assays.

### *E. coli* B834 (DE3) assay
The assay was prepared in 96-well plates (Greiner #655180) with the final volume of 200 μl of M9 minimal salts medium devoid of methionine unless indicated. Overnight cultures of *E. coli* B834 (DE3) grown in LB were back-diluted 1:100 into the wells and supplemented with either 2 μl of prepared bacterial lysates (extracts) or vitamin B$_{12}$ standard solutions used for the growth curves. The growth response was recorded over 20 h at 37°C with 300 rpm agitation, with readings taken every 30 min using a SPECTROstar® Nano plate reader (BMG Labtech) in matrix scan mode using a 2×2 scan matrix with 25 flashes per scan point and path length correction of 5.88 mm for 200 μl volume. For blank corrections of optical density readings, control wells containing media without bacteria were included. Methylcobalamin (Thermo Scientific Chemicals, #A11176ME) was used for the vitamin B$_{12}$ standard curve.

### Statistical analysis and visualisation
Data were analysed and visualised using Prism 10 (Version 10.3.0). AUC analysis provides a comprehensive measure of bacterial growth by

**Table 3. Primers used in the study**

| Primer | 5′ – 3′ sequence | Description |
| --- | --- | --- |
| A88 | TTTCTCTTTGCGCTTGCGTTTCTAGCCCTTATGCAGCCTG | Forward primer for *cbiB* upstream fragment generation |
| A89 | TGGAATGTGGCCGTGCTGTAGCTGAAGATGCGCGAGGAAC | Reverse primer for *cbiB* upstream fragment generation |
| A90 | GTTCCTCGCGCATCTTCAGCTACAGCACGGCCACATTCCA | Forward primer for *cbiB* downstream fragment generation |
| A91 | CGCCAAGCGCGCAATTAACCCGAAGGCTTGCCGCTATCAT | Reverse primer for *cbiB* downstream fragment generation |
| A92 | ATGATAGCGGCAAGCCTTCGGGTTAATTGCGCGCTTGGCG | Forward primer for *cbiB* pFOK backbone generation |
| A93 | CAGGCTGCATAAGGGCTAGAAACGCAAGCGCAAAGAGAAA | Reverse primer for *cbiB* pFOK backbone generation |
| A101 | GAGCCAGATGCGCTACTGAA | Forward primer for *cbiB* mutant screening and validation |
| A102 | TCATGGTGGCTTGAGGCAGC | Reverse primer for *cbiB* mutant screening and validation |
| A161 | TTTCTCTTTGCGCTTGCGTTTCGCCAGCACTTCCAAAAAC | Forward primer for *cbiA* upstream fragment generation |
| A162 | TGGCCCTGGCGGGCACCCCCGACTTCTCCGATGCAACCCT | Reverse primer for *cbiA* upstream fragment generation |
| A163 | AGGGTTGCATCGGAGAAGTCGGGGGTGCCCGCCAGGGCCA | Forward primer for *cbiA* downstream fragment generation |
| A164 | CGCCAAGCGCGCAATTAACCGCGCGTTCAGCGCCACGGCC | Reverse primer for *cbiA* downstream fragment generation |
| A165 | GGCCGTGGCGCTGAACGCGCGGTTAATTGCGCGCTTGGCG | Forward primer for *cbiA* pFOK backbone generation |
| A166 | GTTTTTGGAAGTGCTGGCGAAACGCAAGCGCAAAGAGAAA | Reverse primer for *cbiA* pFOK backbone generation |
| A338 | ACAGCCGGATCATTTGAGCT | Forward primer for *cbiA* mutant screening and validation |
| A339 | CTGTTCCAGCGCTTCTCGCA | Reverse primer for *cbiA* mutant screening and validation |

integrating OD600 readings over time, capturing the full dynamics of the growth curve – including lag, exponential, and stationary phases. Unlike single-point measurements, AUC reflects total biomass accumulation and is less affected by transient fluctuations or noise in the data. With high-resolution measurements taken every 30 min over 20 h, AUC was used as a robust and quantitative way to compare overall growth performance across strains or treatment conditions, especially when differences are subtle or affect growth kinetics rather than final density. To assess statistical significance between groups, we used one-way ANOVA followed by Holm-Šidák multiple comparisons testing, which controls for type I error while maintaining statistical power across multiple pairwise comparisons, with specific details described in the figure legends.

## Acknowledgements
We would like to thank UEA School of Biological Sciences technicians and the admin team for their support throughout the project and Associate Professor Andrew Gates for useful discussions. We thank the *Caenorhabditis* Genetics Centre (CGC), funded by the NIH Office of Research Infrastructure Programs (P40 OD010440). We thank WormBase for providing access to essential *C. elegans* resources.

## Competing interests
The authors declare no competing or financial interests.

## Author contributions
Conceptualization: K.H., A.A.; Data curation: K.H.; Formal analysis: K.H.; Funding acquisition: M.J.S., A.A.; Investigation: K.H., M.J.S., A.A.; Methodology: K.H., M.S., A.A.; Project administration: A.A.; Resources: A.A.; Validation: K.H., A.A.; Writing – original draft: K.H., A.A.; Writing – review & editing: K.H., M.J.S., A.A.

## Funding
This work was supported by a UK Research and Innovation Future Leaders Fellowship [MR/S033769/1 and MR/X024261/1] awarded to A.A. from the Medical Research Council, a Springboard Award from the Academy of Medical Sciences [SBF009\1005] and a Royal Society research grant [RGS\R1\231151] awarded to M.J.S. K.H. was funded by the University of East Anglia doctoral training programme. Open Access funding provided by University of East Anglia School of Biological Sciences. Deposited in PMC for immediate release.

## Data and resource availability
All data generated or analysed during this study are included in this published article and its supplementary information files.

## Peer review history
The peer review history is available online at https://journals.biologists.com/bio/lookup/doi/10.1242/bio.062017.reviewer-comments.pdf

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
