## [Peer Review File · Biology Open]

A simple, fast and inexpensive approach using *E. coli* to detect and estimate vitamin B12 content in microbial extracts

Katarzyna Hencel, Matthew Sullivan and Alper Akay

DOI: 10.1242/bio.062017

Editor: Christopher A. Maher

Review timeline

Original submission:	8 April 2025
Editorial decision:	4 June 2025
First revision received:	15 July 2025
Accepted:	2 August 2025

Original submission

First decision letter

MS ID#: bio.062017

MS Title: A simple, fast and inexpensive approach using *E. coli* to detect and estimate vitamin B12 content in microbial extracts

Authors: Katarzyna Hencel, Matthew Sullivan and Alper Akay

I have now reached a decision on the above manuscript.

The reviewer reports are shown at the bottom of this email or can be accessed, together with a copy of this decision letter, by going to:

As you will see, the reviewers raised a number of substantial criticisms that prevent me from accepting the paper at this stage.

They suggest, however, that a revised version might prove acceptable, if you can address their concerns. If you think that you can deal satisfactorily with the criticisms on revision, I would be pleased to see a revised manuscript. We would then return it to the reviewers.

At this stage, we also ask you to ensure your manuscript complies with our formatting guidelines. Provided you are able to fully address the referees' comments, we are positive about publication of your paper (we accept over 95% of revision submissions) and therefore hope you won't mind any extra work involved in reformatting your manuscript at this point.

Please ensure that you clearly highlight all changes made in the revised manuscript. Please avoid using 'Tracked changes' in Word files as these are lost in PDF conversion.

I should be grateful if you would also provide a point-by-point response detailing how you have dealt with the points raised by the reviewers in the 'Response to Reviewers' box. Please attend to all of the reviewers' comments. If you do not agree with any of their criticisms or suggestions please explain clearly why this is so.

Reviewer 1

Comments for the author

This article presents a new, simple and promising method to measure vitamin B12, that is partially validated in a *C. elegans* research context, generating new useful insight into metabolic interactions between *C. elegans* and its gut microbiota.

VitB12 deficiency has been a topical issue for decades owing to its impact on human health, and the role of host-gut microbes interactions in vitB12 host metabolism is thus of particular interest. It is also of strong interest for the *C. elegans* research community, as the influence of diet and gut microbiota metabolism on host phenotypes has become a chief concern when interpreting the results from *C. elegans* studies on human disorders, innate immunity, stress resistance, toxicology and ageing, with vitB12 metabolism being a particular point of focus. The availability of an accessible and affordable assay to measure vitB12 in biological samples is thus a very welcome addition to the range of more exclusive or expensive existing approaches involving more difficult microorganisms and chromatography techniques.

General comments.

The article is clearly written and easy to follow, providing sufficient details to reproduce its results and implement the approach in one's laboratory.

The methodology is generally convincing, at proof of principle stage, with the findings of the paper indicating that the technique is sensitive and appears quantitative enough over a range of concentrations (0.025-0.4nM or possibly up to 1nM) to enable estimating vitB12 concentrations in suitably prepared microbial samples.

However, (1) a direct side-by-side comparison with another common microorganism-based technique would be more compelling (with standard curves including technical replicate spread for a range of vitB12 concentrations), and (2) a validation of the absolute quantities estimated in biological samples should be performed by a chromatography technique. While the measurement of AUCs for various *C. elegans* gut microbiotas clearly separates vitB12-producing from non producing microbes, it is difficult to determine the robustness of the assay for quantitation purposes (relative or absolute).

Overall, it is a very appealing idea and an interesting approach, but it lacks the depth of information required beyond proof of principle to warrant widespread adoption.

Short of addressing points (1) and (2) it is thus essential that no result is overstated and that the discussion explores all limitations of the technique in its current form (need for further validation, potential for quantisation not fully demonstrated, discussion of potential confounding effects related to differences in methionine and cobalt metabolism across species, short linear range and need for more complex curve fitting over larger ranges of concentrations, identification of hard limits for applicability, etc.). It is also important to state there that the method is not ready as is for wider adoption as a replacement of existing validated approaches. This must also be reflected in the title so that other researchers do not blindly use the technique as a new standard in lieu of already established quantitative techniques. In that respect, a more appropriate title could be: "A simpler, faster and inexpensive approach using *E. coli* to detect and estimate vitamin B12 content in microbial extracts"

Lastly, the samples tested were all extract from environmental microbes grown in aerobic and non fastidious conditions (on simple media). To demonstrate a potentially broader applicability of the approach, other biological extracts (mammalian tissues, anaerobic microbes, food products) should be measured and the influence of confounding factors (methionine traces etc) evaluated. This will also attract a wider readership for the article, including from industry.

Specific minor comments

The introduction is effective but if space allows, a deeper presentation of vitB12 metabolism across living organisms would be welcome.

A brief referenced and more comprehensive listing of other abiotic methodologies (HPLC-UV, reversed-phase HPLC with gradient elution, ICP-M...) to measure vitB12 with their benefit and

limitations (sensitivity, robustness, cost...) possibly in a table would also be useful to help the reader appreciate better where the proposed new method fits. Rationales, statistical approaches and parameters chosen for data analyses should be listed in the method section.

Figure 2 legends should refer to panel E.

In discussion, the authors may consider discussing confounding factors that potential adopters of the approach should consider when planning such experiments.

The discussion might also mention how *E. coli* growth-based approaches may be applied to measure other metabolites of interest and how microbial assays in general should not be forgotten as a powerful and inexpensive alternative to biochemistry approaches.

Reviewer's Responses to Questions

Experimental quality

Does each figure have the proper controls?

If 'No', please indicate reasons in Comments for Author box below.

Reviewer #1:

- Yes

Reviewer #2:

- No

Were the data analyzed using appropriate statistical tests?

If 'No', please indicate reasons in Comments for Author box below.

Reviewer #1:

- Yes

Reviewer #2:

- Yes

Reproducibility

Were experiments performed using adequate number of biological replicates?

If 'No', please indicate reasons in Comments for Author box below.

Reviewer #1:

- Yes

Reviewer #2:

- No

Does the methods section provide sufficient detail to permit reproducibility?

If 'No', please indicate reasons in Comments for Author box below.

Reviewer #1:

- Yes

Reviewer #2:

- Yes

Completeness

Are the manuscript's conclusions supported by the data?

If 'No', please indicate reasons in Comments for Author box below.

Reviewer #1:

- Yes

Reviewer #2:

- No

Scholarship

Do the authors cite and discuss the merits of data that would argue for and against their conclusion?

If 'No', please indicate reasons in Comments for Author box below.

Reviewer #1:

- Yes

Reviewer #2:

- No

Does the manuscript title & abstract accurately reflect the contents of the manuscript, without hyperbole?

If 'No', please indicate reasons in Comments for Author box below.

Reviewer #1:

- Yes

Reviewer #2:

- No

First revision

Author response to reviewers' comments

Point by Point Response to Comments from the Reviewers:

We thank both reviewers for their time and insightful comments on our manuscript. We were particularly encouraged to see that both reviewers agree our manuscript offers an easy-to-use, cost-effective, and accessible method to detect vitamin B12 in suitable samples.

The main issue both reviewers rightly raise is our use of the term “quantification”. We agree with the reviewers that our method does not provide absolute quantification and is not a complete alternative to analytical methods. As the reviewers agree, our method offers an easy alternative to detect vitamin B12 and provides an estimation between samples or between samples and standards within a small linear range. We have updated the manuscript to better reflect these points.

Reviewer 1

- 1- “(1) a direct side-by-side comparison with another common microorganism-based technique would be more compelling (with standard curves including technical replicate spread for a range of vitB12 concentrations), and (2) a validation of the absolute quantities estimated in biological samples should be performed by a chromatography technique. While the measurement of AUCs for various *C. elegans* gut microbiotas clearly separates vitB12-producing from non-producing microbes, it is difficult to determine the robustness of the assay for quantitation purposes (relative or absolute)..... Short of addressing points (1) and (2) it is thus essential that no result is overstated and that the discussion explores all limitations of the technique in its current form”

We agree with the reviewer that additional comparisons and validations are important, and we have revised our manuscript to ensure no results are overstated (see below). However, we do not have access to other strains used in microorganism-based techniques such as colistin-resistant isolate *Lactobacillus leichmannii* NCIB 12519 (cite Pubmed ID 1856292) and do not have the facilities or resources to complete a thorough quantitative chromatographic comparison that the reviewer has suggested, which we believe would be beyond the scope of this work.

We included the following sentence in the discussion to caution these points: “However, we did not test a wider range of culture conditions, such as vitamin B₁₂ concentrations above 1 nM, but as highlighted in this work, we predict such levels would be sufficient for growth of the *E. coli* strain used in the present study. In addition, we have not performed a side-by-side comparison with other bacterial isolates used previously for microbiological assays of vitamin B12 (cite PMID 1856292). Therefore, in future work, we recommend additional testing of culture conditions and using a known vitamin B12-deficient strain as controls. In addition, the range of relative quantification is between 50 and 200 picomolar, which is small and should be used with caution, noting that samples may require dilution to be within this range. For absolute quantification of vitamin B12, analytical methods using liquid chromatography and mass spectrometry should be considered as the validated approach. “

We believe our results are robust, based on reproducibility across multiple biological replicates, and the approach was sufficient to identify the limit of detection for vitamin B12 by the use of known vitamin B12 concentrations prepared from standards. The standard curve (See below) is highly reproducible, though within a short linear range, as mentioned by the reviewer, thus requiring the application of a dilution series in order to bring unknown samples to within this window for accurate estimation of quantity. In addition, we were able to use our microbiological assay to demonstrate that cell-free extracts from multiple non-B12-producing bacteria of *C. elegans* gut microbiota origin, and mutants *C. aquatica* defective for B12-production, do not produce B12, using our *E. coli*-based approach.

For the benefit of the reviewers we have produced a standard curve using data plotted in Figure 2B, which shows good fit of the data in this simple linear regression analysis from AUC data between 0 - 0.2 nM B12 (see Figure 1). We have added this linear model to Figure 2F.

Figure 1. Area under the curve analysis of *E. coli* B834 cultures supplemented with vitamin B₁₂ concentrations prepared from known standards. The line represents simple linear regression and the equation and R² value are shown from three independent experiments.

- 2- “it is thus essential that no result is overstated and that the discussion explores all limitations of the technique in its current form (need for further validation, potential for quantisation not fully demonstrated, discussion of potential confounding effects related to differences in methionine and cobalt metabolism across species, short linear range and need for more complex curve fitting over larger ranges of concentrations, identification of hard limits for applicability, etc.). It is also important to state there that the method is not ready as is for wider adoption as a replacement of existing validated approaches. This must also be reflected in the title so that other researchers do not blindly use the technique as a new standard in lieu of already established quantitative techniques. In that respect, a more appropriate title could be: “A simpler, faster and inexpensive approach using *E. coli* to detect and estimate vitamin B12 content in microbial extracts”

We have edited the manuscript so that we use “estimate” instead of “quantification” throughout the manuscript. We have revised the manuscript title to avoid overstating the findings as the reviewer has suggested. We have added sections on validation and quantitation limits as described in response to comment 1. Our original manuscript already included a discussion on potential confounding effects of methionine, and our results (400 nM methionine vs 20 pM B12) are similar to prior studies that show that methionine is required at orders of magnitude more than B12 in order to rescue growth in auxotrophic mutants of *E. coli* (Chiao et al 1950 #15 in manuscript). We have added an additional sentence to the discussion to state that, although we did not analyse different concentrations of methionine, it is essential to use appropriate controls. We believe our method is ready for wider adoption due to its simplicity and cost-effectiveness, but it should be used within its limitations. Since the revised manuscript focuses on the detection and estimation of vitamin B12 rather than quantification, we believe this concern is now addressed.

- 3- “Lastly, the samples tested were all extract from environmental microbes grown in aerobic and non fastidious conditions (on simple media). To demonstrate a potentially broader applicability of the approach, other biological extracts (mammalian tissues, anaerobic microbes, food products) should be measured and the influence of confounding factors (methionine traces etc) evaluated. This will also attract a wider readership for the article, including from industry.”

We agree with the reviewer that demonstrating the broader applicability of the method using other biological materials is important, but we suggest this is beyond the scope of this

work and would be the topic of future studies, as we have alluded to in the discussion. As we are a *C. elegans* lab, our main focus was to develop a method to investigate B12 production in the microbiome of nematodes. However, in the future, we plan to adapt our method to different biological materials, including nematodes themselves, mammalian tissues, and possibly food material. We believe that pursuing these extensions is beyond the scope of this manuscript at this stage.

Minor comments

- 1- “The introduction is effective but if space allows, a deeper presentation of vitB12 metabolism across living organisms would be welcome.”
We included a brief section on the role of B12 in *C. elegans* development. Our manuscript focuses more on its detection rather than its function.
- 2- “A brief referenced and more comprehensive listing of other abiotic methodologies (HPLC-UV, reversed-phase HPLC with gradient elution, ICP-MS...) to measure vitB12 with their benefit and limitations (sensitivity, robustness, cost...) possibly in a table would also be useful to help the reader appreciate better where the proposed new method fits.”
We added a section in the introduction where we describe chromatography and mass spectrometry methods in more detail. We believe that any further comparisons would be more appropriate for a review paper.
- 3- “Rationales, statistical approaches and parameters chosen for data analyses should be listed in the method section.”
We added a new paragraph to this effect in the end of the methods section titled ‘Statistical analysis and visualisation’
- 4- “Figure 2 legends should refer to panel E.”
This has now been resolved.
- 5- “In discussion, the authors may consider discussing confounding factors that potential adopters of the approach should consider when planning such experiments.”
We have included this in the updated discussion, particularly emphasising the use of proper controls to account for potential confounding effects from methionine and employing chromatography techniques for absolute quantification.
- 6- “The discussion might also mention how *E. coli* growth-based approaches may be applied to measure other metabolites of interest and how microbial assays in general should not be forgotten as a powerful and inexpensive alternative to biochemistry approaches.”
We agree with this reviewer entirely and have added a sentence to reflect this at the end of our discussion.

Reviewer 2

- 1- “* The manuscript entitled High-throughput detection and quantification of vitamin B12 in microbiome isolates using *Escherichia coli* is a well written manuscript with detailed description.
* As the paper explains, *E. coli* B834 (DE3) is a methionine auxotroph, meaning it requires methionine for growth. The assay relies on the growth of *E. coli* B834 (DE3) in response to the presence of vitamin B₁₂ in the sample.
* How authors have described the B12 is being produced by different *C. elegans* microbiome isolates is appreciated. However, currently, in this article, the terms assay or detection would be more precise than quantification, as this method is sufficient to reveal differences in B12 production by different bacteria, but not quantitating the level present. More direct analytical methods may be needed to achieve highly reliable and precise B12 quantitative measurements.
* There is over interpretation about the quantification of the B12 in the given sample.
* Author can consider changing the title as well by replacing the quantitation with assessment.”
We agree with Reviewer 2 and have updated the manuscript and its title to better reflect the capability of our method (in line with Reviewer 1’s similar comments). We have added

additional sections in the introduction and discussion to emphasise chromatography approaches for absolute quantification and to illustrate that our method is more suitable for estimating B12 levels throughout the manuscript.

Minor comments

- 1- “Page-3, lane 38 add *L. leichamnii* in bracket first time and also for other strains mentioned in manuscript e.g. page 4 lane 1 etc. lane 53 replace typhimurium with Typhimurium Page-16, lane 1 *E. coli* - *E. coli*”
We have addressed these issues in the updated manuscript.
- 2- “As the author explains, *E. coli* B834 (DE3) a methionine auxotroph, meaning it requires methionine for growth. The assay relies on the growth of *E. coli* B834 (DE3) in response to the presence of vitamin B₁₂ in the sample, and *E. coli* B834 (DE3) cells could grow in response to the methionine present, rather than solely in response to the vitamin B₁₂ content. So, a control experiment that can differentiate between growth due to vitamin B₁₂ and growth due to exogenous methionine would be clear for methionine interference.”
The potential confounding effects of methionine were already mentioned in the manuscript, and our results include such controls. For example, Figure 1A shows that 400nM methionine and 1nM B12 support similar growth of the B834 strain in M9, a chemically defined minimal medium to which we either added or withheld methionine. In Figure 2C we demonstrated that bacterial extracts prepared from the same culture density (1 mL of 1.0 OD_{600nm}; approximately 5x10⁸ bacterial cells of biomass) from *C. aquatica* B12- *cbiA*::*cbiB*-mutants do not support any growth of *E. coli* B834. Conversely, extracts from the same volume/number of WT *C. aquatica* DA1877 cells enabled robust growth. Given that equivalent biomass was used to prepare the extracts, these experiments indicate that methionine contained within the cell-free bacterial extracts is not sufficient to support B834 growth. Figure 3 reveals that bacteria lacking the B12 synthesis pathway similarly do not support B834 growth, suggesting that under these conditions, methionine is not a confounding factor. However, we have added a section in the discussion noting that at different quantities of material used, this could be problematic, and it remains important to use appropriate controls.
- 3- “Authors can put bit deeper explanation in the discussion section.”
We included a section on confounding effects and the proper application of the method.
- 4- “Conclusion section lane 2, also author have emphasised on the quantifying the B12 level, but there is no quantification involved in the sample, just the indication of the presences of B12 can be monitored by using the *E. coli* strain”
The conclusion section has now been merged with the discussion, and we have replaced quantification with estimation throughout the manuscript.

Second decision letter

MS ID#: bio.062017R1

MS Title: A simple, fast and inexpensive approach using *E. coli* to detect and estimate vitamin B12 content in microbial extracts

Authors: Katarzyna Hencel, Matthew Sullivan and Alper Akay

I am happy to tell you that your manuscript has been accepted for publication in Biology Open, pending our standard publication integrity checks. It was accepted on 2nd August 2025.

Reviewer 1

Comments for the author

The authors have addressed all reviewer comments in a satisfactory manner and as such the manuscript is now ready for publication (I did not spot check for grammar or spelling errors though).